# Using Chemometric Analyses for Tracing the Regional Origin of Multifloral Honeys of Montenegro

**DOI:** 10.3390/foods9020210

**Published:** 2020-02-18

**Authors:** Vesna Vukašinović-Pešić, Nada Blagojević, Snežana Brašanac-Vukanović, Ana Savić, Vladimir Pešić

**Affiliations:** 1Faculty of Metallurgy and Technology, University of Montenegro, 81000 Podgorica, Montenegro; nadab@ucg.ac.me (N.B.); snezana.v@ucg.ac.me (S.B.-V.); 2Department of Biology and Ecology, Faculty of Sciences and Mathematics, University of Niš, 18000 Niš, Serbia; anka@pmf.ni.ac.rs; 3Department of Biology, University of Montenegro, 81000 Podgorica, Montenegro; vladopesic@gmail.com

**Keywords:** honey, regional origin, chemometric analysis, mineral content, Montenegro

## Abstract

This is the first study of mineral content and basic physicochemical parameters of honeys of Montenegro. We examined honey samples from eight different micro-regions of Montenegro, and the results confirm that, with the exception of cadmium in samples from two regions exposed to industrial pollution, none of the 12 elements analyzed exceeded the maximum allowable level. The samples from areas exposed to industrial pollution were clearly distinguished from samples from other regions of Montenegro in the detectable contents of Pb, Cd, and Sr. This study showed that chemometric techniques might enhance the classification of Montenegrin honeys according to their micro-regional origin using the mineral content. Linear discriminant analysis revealed that the classification rate was 79.2% using the cross-validation method.

## 1. Introduction

Honey is a complex natural product, whose characteristics depend on the flower nectar from which it is obtained, but also on other factors such as geographical origin, bee species, season, type of processing and storage [1]. It is known that pollution and a number of different pollutants present in its foraging areas have an impact on honeybees [2] but also on nectar-providing plant species. Therefore, it is necessary to assure geographical traceability and determine the botanical origin of the foraging area of the beehive.

As stated by Karabagias and Karabournioti [3], the authentication of honey is gaining in importance and includes a number of contending parties from producers and sellers to consumers and control labs. A number of papers have shown that specific physicochemical parameters and mineral contents in combination with chemometric analyses can be a useful tool in discovering botanical and/or geographical origin of honeys that may enter the market [1,3,4].

Tracing the geographical origin of honeys can provide important information about the potential contamination of the area from which the honey production material comes. Therefore, ensuring high standards in terms of product safety leads to the need to examine the contents of essential and toxic elements in honey. Due to its bioaccumulation ability, honey can be used as an indicator of metal pollution, especially of toxic pollutants such as Pb, Cd, and As [5,6,7].

Due to its geographical position, climate conditions and richness of the nectar-providing plants, Montenegro provides favorable natural conditions for more intensive development of beekeeping. According to the data for 2011, the population in Montenegro was 625,266, while the honey production for that year was 394 t, and the average annual consumption 1.2 kg per person, meaning that a large part of honey consumption in Montenegro is imported [8]. Data for the last few years show an increase in honey production (627 t for 2016) but also an increase in the average annual consumption of honey per person (2.76 kg) [9].

The majority of honey on the market in Montenegro are multifloral (derived from a large number of nectar-providing plant species in the honeybees foraging area). Most of these honey types are recognizable by their local or regional origin (e.g., Katunski med (= honey), Pivski med, Piperski med.). It is worth mentioning that Montenegro and its regions are known to harbor a high number of regional floral endemics [10] that likely affect the composition and properties of honey.

There is a lack of information on the mineral content and basic physicochemical parameters of honey from the territory of Montenegro. Moreover, there is no continuity in monitoring the quality of honey, especially in areas that are exposed to the effects of potential pollution sources. Due to the high consumption of local honey in the diet, the need and obligation for its systematic characterization are highly required.

This study is aimed to investigate the mineral content and the basic physicochemical parameters of honeys from different micro-regions of Montenegro. We evaluated the usefulness of chemometric analyses for the classification of honeys according to its regional origin.

## 2. Materials and Methods

Twenty-four honey samples as indicated in Figure 1 were collected from eight micro-regions of Montenegro, i.e., (1) Piva, (2) Zbljevo, (3) Potrlica, (4) Mijakovići, (5) Piperi, (6) Martinići, (7) Katunska, and (8) Zeta. The Piva, Zbljevo, Potrlica and Mijakovići micro-regions are situated in the continental part of Montenegro (Alpine biogeographical region, see Figure 1) while the four other micro-regions are situated in the sub-Mediterranean part of the country belonging to the Mediterranean biogeographical region [10]. The climate in the latter region is mainly Mediterranean-Adriatic with relatively dry and warm summers (the average air temperature of the warmest month > 20 °C), but humid and mild winters (the average air temperatures varies from 6 to 9 °C), while the Alpine region has a “continental” type climate, with relatively cool and humid summers and long and harsh winters [10].

Samples were taken from individual beekeepers during the harvesting season 2015. All samples were multifloral as confirmed by the suppliers. The samples were stored in glass flasks at room temperature before analysis. Physicochemical parameters (pH, electrical conductivity (EC), free acidity (FA) and moisture) were analyzed using the Harmonized Methods of the International Honey Commission [11].

The mineral composition of honey was analyzed by inductively coupled plasma-optical emission spectrometry (ICP-OES). About 1 g of each honey sample was digested with 14 mL 65% HNO_3_ and 2 mL 35% H_2_O_2_ on a hot plate to near dryness. The sample containing a volumetric flask was cooled at room temperature before the addition of deionized water to the mark on the flask. All samples were prepared in triplicate and their average value was assessed.

The concentration of twelve elements (Pb, Cd, Cu, Zn, Fe, Cr, Sr, Ba, Ca, Na, K, Mg) were determined by ICP-OES according to the iCAP 6000 spectrometer method.

All statistical analyses were performed using SPSS 17.0 (SPSS Statistics for Windows, Version 17.0. SPSS Inc., Chicago, IL, USA). Data were expressed as mean ± standard deviation. A Kolmogorov–Smirnov test showed that all analyzed physicochemical parameters were normally distributed, while the content of Pb, Cd, Sr and Ba in some regions exhibited significant differences from the normal distribution. The one-way analysis of variance (ANOVA) was performed on physicochemical parameters in order to determine if there any significant differences between studied micro-regions at the confidence level 0.05. The Kruskal–Wallis test was used to investigate whether the mineral contents varied significantly between the investigated micro-regions. The relationship between the mineral content and physicochemical parameters were analyzed using the Spearman’s correlation analysis. For checking similarities between samples of honey of different geographical origin we used two chemometric analyses: PCA and LDA. Statistical methods based on principal component analysis (PCA) and linear discriminant analysis (LDA) have been used. The LDA was performed using R.3.5.3, while the PCA was made by using MVSP version 3.21.

## 3. Results

The mineral content of honey samples from different geographical areas of Montenegro is presented in Table 1. The value presented for each element is the average concentration observed. A significant difference has been observed in the concentrations of Pb, Cd and Sr (*p* = 0.002) between studied micro-regions. In most analyzed samples the concentrations of above-listed elements were below the limit of detection except in the samples from Potrlica, Zbljevo and Mijakovići. The highest Cd concentration was observed in samples from Potrlica (0.08 ± 0.01 mg/kg). The highest concentration of Pb (0.21 ± 0.06 mg/kg) and Sr (0.12 ± 0.00 mg/kg) were recorded in samples from Zbljevo.

The concentrations of examined physicochemical parameters of honey samples are given in Table 1. The moisture level had similar values across studied regions and ranged from 14.92 ± 0.78% (Piperi) to 16.22 ± 0.36% (Zeta). The pH of studied honey samples varies between 3.87 and 4.49 and was lowest in samples from the Piva region (3.87 ± 0.36) and highest in honey samples from Mijakovići (4.49 ± 0.14). A significant difference has been observed in pH according to the honey regional origin (*p* = 0.048). The electrical conductivity varied from 0.39 to 0.93 mS/cm and was lowest in samples from Katunska (0.39 ± 0.08 mS/cm) and highest in honey samples from Zeta (0.93 ± 0.15 mS/cm). A significant difference has been observed in electrical conductivity according to the honey geographical origin (*p* = 0.013). Free acidity varied from 25.00 to 41.67 meq/kg and was lowest in samples from Katunska (25.00 ± 7.21 meq/kg) and highest in honey samples from the Piva region (41.67 ± 12.10 meq/kg).

Correlation analysis revealed significant correlation between contents of K (R = 0.800, significance < 0.001) and Mg (R = 0.758, significance < 0.001) from one side and pH from the other one (Table 2).

The first principal component explains 42.53% of the total variability and is mostly determined by Cd (R = 0.416), Pb (R = 0.398), and Cu (R = 0.352). The PC2 explains 17.97% and is mostly determined by Mg (R = −0.577), K (R = −0.54), and Na (R = −0.425). Mutual projections of factor scores and their loadings for the first two PCs are presented in Figure 2. As can be seen from the projection plot the separation of the analyzed honey samples is much clearer along the *X*-axis. On the one side, there are localities Piva, Piperi, Katunska, Zeta and Martinići in whose honey samples Cd, Pb, and Sr were not detected. On the other side, there are Mijakovići, Potrlica and especially Zbljevo, whose honey samples concentrations of Cd, Pb and Sr were detected.

LDA performed on the geographical origin revealed that the cross-validation classification was correct for 79.17% of samples (Table 3). The smallest percent of good classification was achieved in the case of honey samples from Katunska, while the highest in the case of honeys from Piperi, Zeta, Mijakovići, and Zbljevo.

The bidimensional plot (Figure 3) of the first two functions show four distinct clusters, three of them corresponding to Mijakovići, Potrlica, and Zbljevo regions, while all other regions were clustered together. The first discriminant function explains 94.2% of the total variance and it is dominated by Cd content (R = 0.95). The second discriminant function explains 4.4% of the total variance and is dominated by Pb (R = −0.55) and Sr (R = −0.59) contents.

## 4. Discussion

The values of the mineral contents have been compared with those established by the EU regulations [12]. With the exception of the concentrations of cadmium in samples from Zbljevo and Potrlica, none of the 12 elements analyzed exceeded the maximum allowable level established by the EU regulations. Our results revealed significant differences in the concentrations of Pb, Cd, and Sr between the studied geographical areas of Montenegro. The latter elements were detected only in the samples from Mijakovići, Potrlica and Zbljevo regions, which are likely under the influence of the Pljevlja Thermal Power Plant (Zbljevo, and in less extent Mijakovići) and the Pljevlja coalmine (Potrlica).

The Pb content in the analyzed honey samples varied from 80–210 μg/kg. These values were lower in comparison with the honey from Serbia (290 μg/kg [13]) and Italy (289 μg/kg [14]) but higher in comparison with those from Croatia (5.43–11.3 μg/kg [15]) and Bosnia and Hercegovina (13.4 μg/kg [16]). All these values are below the maximum allowable level established by the EU regulations (0.5 mg/kg) [12].

The values reported for Cd in this study (20–80 μg/kg) were higher in comparison with the honey from Croatia (0.69–12.8 μg/kg [15]), Bosnia and Hercegovina (0.013–22.9 μg/kg [16]), Romania (0.5–11.60 μg/kg [17]), Italy (8–18 μg/kg [14]), Spain (0.7–50 μg/kg [17]) and Serbia (0.59–30 μg/kg [1,13]). The values of Cd content in the samples from Potrlica and Zbljevo exceeds the maximum allowable level established by the EU legislation (0.05 mg/kg) [12]. As the main sources of Cd are recognized as the presence in sewage sludge and smelting from the nearby Pljevlja Thermal Power Plant (Zbljevo), or mining from the Pljevlja coalmine (Potrlica).

The Sr content in honey samples from our study varied from 0.07–0.12 μg/kg and was in the same range as those from Serbia (0.09–0.19 μg/kg [13]).

The most abundant element in honey samples was kalium, followed by Ca, Mg, Na, and Fe. In our study, we found that the content of kalium and magnesium correlated with pH. The average levels of K content ranged from 713–2589.33 mg/kg and was in the same range with those from Croatia (304.7–2824.4 mg/kg [15]), but lower in comparison with the maximum values established for honeys from Bosnia and Herzegovina (14.81–4895.73 μg/kg [16]). On the other hand, the range of concentrations of kalium in the honey from Serbia (400–1755 mg/kg [1,13]) and Slovenia (1090–1220 mg/kg [18]) were lower. The Mg content in honey samples from our study varied from 29.52 to 76.33 mg/kg. In neighboring countries the Mg content in honey varied in a similar range, as: 28.83 to 101.50 mg/kg [13] in Serbia, 2.18 to 166.04 mg/kg [16] in Bosnia and Herzegovina and from 8.02 to 59.1 mg/kg [16,19] in Croatia.

In our study, we used two chemometric analyses, PCA and LDA, respectively to test similarities between honey samples of honey of different geographical origins. Both applied methods separated the regions exposed to industrial pollution (Mijakovići, Potrlica, and Zbljevo) which are characterized by detectable content of Cd, Pb and Sr in their honey samples.

Using LDA it’s possible to evaluate the capacity to correctly predict the group to which the unknown samples belong. In our study LDA analysis performed on the geographical origin revealed that the cross-validation classification was correct for 79.17% of the samples. The obtained values are in the range for those from Serbia (Zlatibor: 94.73%, Vojvodina: 70.58% [1]). On the other hand, our value was greater than those reported in the case of Romania honeys where only 21.2% were correctly classified according to their geographical origin [4].

The smallest percentage of good classification was achieved in the case of honeys from Katunska. Of the three samples from the latter region, only one was correctly classified, while the other two being misclassified as Piperi and Zbljevo, respectively. It is known that large numbers of beekeepers (especially from Katunska) in a part of the year (most often in summertime) move their bee colonies to geographically distant areas. On the other hand, the highest percentage of good classification was achieved in the case of honeys from Piperi, Zeta, Mijakovići, and Zbljevo. One cause may be that most of these sites (i.e., Zeta, Mijakovići, and Zbljevo) are more exposed to industrial pollution, resulting in increased concentration of heavy metals (Pb, Cd, and Sr showing significant difference (*p* < 0.05) between studied regions) in their honeys, which, in turn, increase the success rate of the classification of honey according to their geographical origin.

## Figures and Tables

**Figure 1 foods-09-00210-f001:**
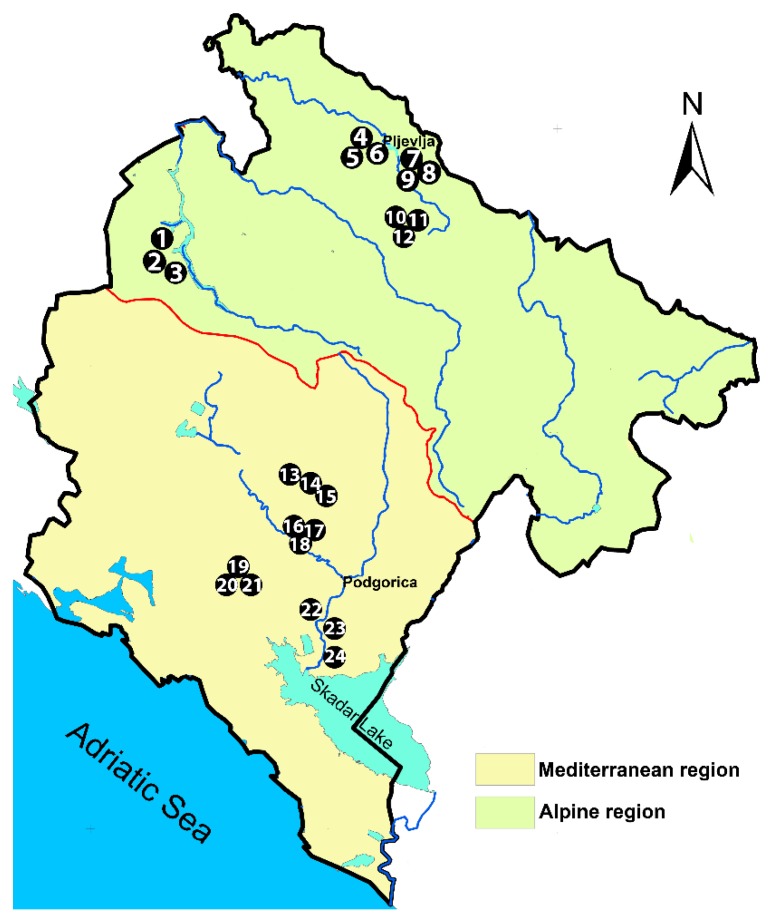
Map of Montenegro with marked locations of honey sampling from eight micro-regions (in parentheses are given sampling location numbers): Piva (1–3), Zbljevo (4–6), Potrlica (7–9), Mijakovići (10–12), Piperi (13–15), Martinići (16–18), Katunska (19–21), and Zeta (22–24).

**Figure 2 foods-09-00210-f002:**
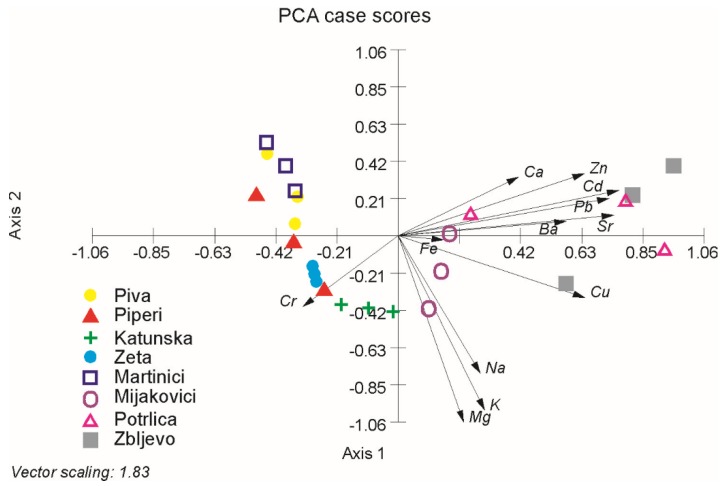
Principal component analysis (PCA) of the mineral content scores of analyzed Montenegrin honey samples.

**Figure 3 foods-09-00210-f003:**
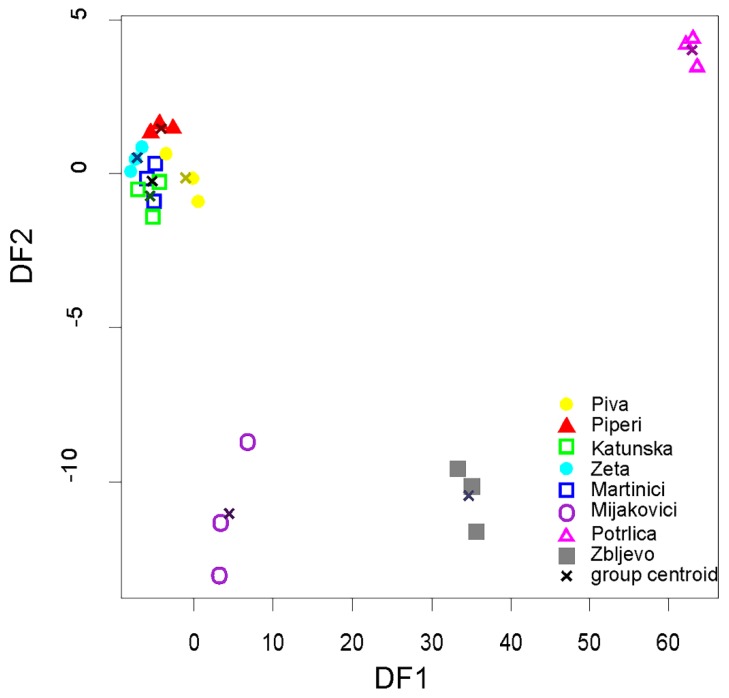
Linear discriminant score plot of analyzed honey samples.

**Table 1 foods-09-00210-t001:** Mineral content and physicochemical parameters of honey samples from studied micro-regions of Montenegro.

Regions		Cu mg/kg	Zn mg/kg	Fe mg/kg	Cr mg/kg	Ca mg/kg	Na mg/kg	K mg/kg	Mg mg/kg	Pb mg/kg	Cd mg/kg	Sr mg/kg	Ba mg/kg	EC mS/cm	pH	Moisture %	Free Acidity meq/kg
Katunska	Mean	0.78	1.61	13.00	0.90	100.49	53.62	2155.00	76.33	ND *	ND	ND	ND	0.39	4.43	16.15	25.00
	SD	0.15	0.01	1.00	0.50	3.35	6.58	115.00	14.74					0.08	0.08	0.37	7.21
Martinići	Mean	0.31	1.18	7.67	0.30	109.82	34.61	713.00	31.38	ND	ND	ND	ND	0.82	3.95	15.40	29.33
	SD	0.09	0.60	1.53	0.10	1.36	2.24	252.00	11.89					0.04	0.17	1.16	3.51
Mijakovići	Mean	0.62	1.40	3.95	0.10	80.50	46.00	2589.33	68.77	0.08	0.02	0.07	0.27	0.86	4.49	16.04	30.00
	SD	0.08	0.00	1.05	0.01	29.50	4.00	652.50	9.75	0.01	0.00	0.01	0.11	0.09	0.14	0.49	3.61
Piperi	Mean	0.53	0.84	14.67	1.13	76.70	39.93	1760.00	37.91	ND	ND	ND	ND	0.75	4.48	14.92	28.00
	SD	0.18	0.19	2.52	0.15	3.04	5.72	685.00	12.12					0.11	0.35	0.78	2.00
Piva	Mean	0.53	0.52	7.33	0.27	74.65	39.76	844.33	29.52	ND	ND	ND	ND	0.44	3.87	15.36	41.67
	SD	0.17	0.16	1.53	0.15	22.46	8.62	333.78	12.31					0.21	0.36	0.44	12.10
Potrlica	Mean	0.87	3.00	7.53	0.12	152.00	46.00	1945.33	61.83	0.12	0.08	0.08	1.44	0.68	4.32	16.05	37.00
	SD	0.22	0.61	6.05	0.07	84.26	8.54	189.72	9.22	0.02	0.01	0.05	1.67	0.20	0.19	1.43	11.14
Zbljevo	Mean	0.98	4.13	15.93	0.26	103.00	56.67	1651.33	41.00	0.21	0.07	0.12	0.62	0.63	4.11	15.40	36.33
	SD	0.19	3.30	5.71	0.06	10.15	3.51	1005.47	17.78	0.06	0.01	0.00	0.34	0.32	0.36	0.14	3.51
Zeta	Mean	0.47	0.93	11.00	0.47	48.88	63.94	1285.00	57.23	ND	ND	ND	ND	0.93	4.34	16.22	32.00
	SD	0.02	0.29	0.00	0.07	2.20	3.46	135.00	0.17					0.15	0.23	0.36	12.12
Total	Mean	0.64	1.64	10.14	0.44	93.25	47.57	1617.92	50.50	0.05	0.02	0.03	0.29	0.69	4.25	15.69	32.42
	SD	0.25	1.59	4.80	0.40	40.15	10.58	750.73	19.77	0.08	0.03	0.05	0.70	0.24	0.31	0.79	8.49
F ^a^/χ2 ^b^		6.59 ^a^	3.17 ^a^	5.08 ^a^	10.65 ^a^	2.64 ^a^	8.68 ^a^	4.61 ^a^	6.63 ^a^	22.87 ^b^	22.57 ^b^	22.06 ^b^	21.62 ^b^	3.79 ^a^	2.69 ^a^	1.17 ^a^	1.42 ^a^
significance		0.001 ^c^	0.026 ^c^	0.003 ^c^	0.000 ^c^	0.051 ^c^	0.000 ^c^	0.005 ^c^	0.001 ^c^	0.002 ^d^	0.002 ^d^	0.002 ^d^	0.003 ^d^	0.013 ^c^	0.048 ^c^	0.373 ^c^	0.263 ^c^

* ND = not detected. a = ANOVA F statistics, b = Kruskal-Wallis Chi-Square statistics, c = ANOVA significance, d = Kruskal-Wallis significance.

**Table 2 foods-09-00210-t002:** Results of the correlation analysis between the mineral content and physicochemical parameters of analyzed honey samples from Montenegro.

		Electrical Conductivity	pH	Moisture	Free Acidity
Pb	R	−0.028	−0.052	0.013	0.224
Cd	R	−0.087	−0.030	0.080	0.259
Cu	R	−0.239	0.240	−0.043	−0.029
Zn	R	−0.329	−0.183	0.052	0.035
Fe	R	−0.379	0.044	−0.115	−0.194
Cr	R	−0.179	0.335	−0.069	−0.285
Sr	R	−0.006	0.074	0.009	0.125
Ba	R	−0.064	0.114	−0.275	−0.082
Ca	R	−0.229	−0.019	-0.031	−0.061
Na	R	0.079	0.382	0.214	−0.173
K	R	0.171	0.800 **	0.218	−0.250
Mg	R	0.159	0.758 **	0.325	−0.391

** significance < 0.01.

**Table 3 foods-09-00210-t003:** Classification of honey according to their regional origin using the linear discriminate analysis.

Predicted	Piva	Piperi	Katunska	Zeta	Martinići	Mijakovići	Potrlica	Zbljevo
Piva	66.67%	0.00%	0.00%	0.00%	33.33%	0.00%	0.00%	0.00%
Piperi	0.00%	100.00%	33.33%	0.00%	0.00%	0.00%	0.00%	0.00%
Katunska	0.00%	0.00%	33.33%	0.00%	0.00%	0.00%	0.00%	0.00%
Zeta	0.00%	0.00%	0.00%	100.00%	0.00%	0.00%	0.00%	0.00%
Martinići	33.33%	0.00%	0.00%	0.00%	66.67%	0.00%	0.00%	0.00%
Mijakovići	0.00%	0.00%	0.00%	0.00%	0.00%	100.00%	0.00%	0.00%
Potrlica	0.00%	0.00%	0.00%	0.00%	0.00%	0.00%	66.67%	0.00%
Zbljevo	0.00%	0.00%	33.33%	0.00%	0.00%	0.00%	33.33%	100.00%

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
