# Peer review of "Using Chemometric Analyses for Tracing the Regional Origin of Multifloral Honeys of Montenegro"

_foods, 2020, doi:10.3390/foods9020210_

Round 1

Reviewer 1 Report

It will have being good if authors could sample soils from the farms where these honey were collected and check whether these heavy metal levels are low or high? This could be one of the sources of the heavy metals in the honey samples analyzed.

The climatic conditions, as well as location maps, should be described and presented.

Reviewer 2 Report

Review comments

Paper Title: Using Chemometric Analyses for tracing the regional origin of multifloral honeys of Montenegro

The manuscript refers to the relation of the mineral content, some common physichochemical parameters and geographical origin of honeys from Montenegro.

Considering the honey from Montenegro are poorly studied, this paper can have some scientific interest. Nevertheless, the document needs a considerable improvement, due to this I made some comments about the document in order to its improvement.

The main concern is about the influence of the biogeographical origin in the honey samples. So a biogeographical description of the 8 areas where honeys were collected should be included as this description has interest to understand the differences in honeys samples. It would be very useful a map showing the geographical origin of the 24 samples.

Also, information about the botanical origin of the samples is necessary to understand the differences in the mineral content and the physicochemical parameters.

Other comments:

Line 91. As heavy metals and other elements are shown in table 1, is recommended to delete the word heavy starting the sentence.

Line 93 and following. It is better to understand the differences among the samples to mark these differences in the table. Are the mentioned differences for all the samples or considering each geographical origin?. Where are the results of the Kruskal-Wallis test?

Line 113. Table 2 could be improved indicating the significant correlation above the Coefficient.

Line 118. Some metal elements are not included in the PCA analysis, this should be mentioned and discussed in the text.

Finally, the discussion should be enlarged including information about the botanical origin of the honeys studied and the relation among geographical origin, botanical origin and metal content. This also could explain the differences in honeys from Katunska (lines 190-192).

I hope my comments help authors to improve the paper.

Round 2

Reviewer 2 Report

The paper was changed according the comments

Author Response

The authors would like to thank to anonymous reviewer for positive decision.